# Metabolic syndrome and idiopathic sudden sensori-neural hearing loss

**Massimo Rinaldi[1], Giada Cavallaro[1], Marica Cariello [2], Natasha Scialpi [2], Nicola Quaranta [1]***

**1** Otolaryngology Unit, Department of Biomedical Sciences, Neuroscience and Sensory Organs, University of Bari "Aldo Moro", Bari, Italy, **2** Clinica Medica Cesare Frugoni, Department of Interdisciplinary Medicine, University of Bari Aldo Moro "Aldo Moro", Bari, Italy

* nicolaantonioadolfo.quaranta@uniba.it

## Abstract

The purpose of this study was to evaluate the association between the presence of Metabolic Syndrome (MetS) and idiopathic sudden sensorineural hearing loss (ISSHL) and the impact of MetS on recovery of patients with ISSHL. 39 Patients with ISSHL and 44 controls were enrolled in this study. Demographic, clinical characteristics and hearing recovery were evaluated. MetS was defined according to the diagnostic criteria of International Diabetes Federation (IDF) consensus definition. Patients affected by ISSHL presented a body mass index (BMI), waist circumference, waist hip ratio (WHR), fasting glucose and blood pressure significantly higher compared to controls. Considering patients with central obesity, 5 controls and 15 ISSHL patients met the criteria of MetS. According to Siegel criteria, a complete or partial recovery was observed in 60% of patients with MetS and in 91,66% of patients without MetS. MetS was associated with ISSHL and this association negatively influenced the hearing recovery of these patients.

## Introduction

Idiopathic sudden sensorineural hearing loss (ISSHL) is defined, according to American Academy of Otolaryngology, as a hearing loss of at least 30 dB over 3 contiguous test frequencies occurring within a 72-h period [1]. The incidence of ISSHL is approximately 10/100,000 person per year, with no differences in gender and affected side [2]. Pathogenesis of ISSHL is controversial. Viral infections, vascular occlusions, immune-mediated mechanisms and cochlear membrane breaks have been proposed as pathogenic mechanisms, however clinical and experimental evidence that confirm one or other mechanisms is lacking [3]. Recently, in patients affected by ISSHL, endothelial dysfunction [4,5] and a high prevalence of cardiovascular risk factors [6] have been reported. Prospective studies have shown that patients affected by ISSHL have a higher risk of developing a cerebrovascular accident especially if ISSHL is associated with vertigo [7,8], suggesting that hearing loss may share similar etio-pathogenic mechanisms.

Metabolic syndrome (MetS) is a complex disorder defined by a cluster of interconnected factors that increase the risk of cardiovascular atherosclerotic diseases and diabetes mellitus type 2 [9,10]. The International Diabetes Federation has proposed the diagnostic criteria of

**Data Availability Statement:** Data are available in the Supporting Information files.

**Funding:** The author(s) received no specific funding for this work.

**Competing interests:** The authors have declared that no competing interests exist.

MetS, that is defined by the presence of central obesity (which is measured by waist circumference with gender and ethnicity specific values) and any two of the following: *raised triglycerides* (> 150 mg/dL (1.7 mmol/L), or specific treatment for this lipid abnormality), *reduced* HDL cholesterol (< 40 mg/dL (1.03 mmol/L) in males, < 50 mg/dL (1.29 mmol/L) in females, or specific treatment for this lipid abnormality), *raised blood pressure* (BP) (systolic BP > 130 or diastolic BP >85 mm Hg, or treatment of previously diagnosed hypertension), *raised fasting plasma glucose* (FPG) (>100 mg/dL (5.6 mmol/L), or previously diagnosed type 2 diabetes) [11]. Chien et al [12] in a case control study have reported that MetS increase the risk of sudden hearing loss and that the risk increase with the number of MetS components. In addition the presence of MetS has been associated with poorer hearing recovery by other authors [13–15] that however have not used central obesity as a criterion for MetS.

The aim of the present study was therefore to evaluate the association between the presence of MetS and ISSHL and the influence of MetS on the prognosis of ISSHL.

## Material and methods

### Study population

39 Patients with ISSHL and 44 age and sex matched control were enrolled in this study from January 2017 to November 2018. All patients denied previous episodes of ISSHL and none started the treatment before undergoing hearing and blood tests. Exclusion criteria were hearing loss caused by acoustic neuroma, central lesions, autoimmune disorders, Meniere's disease, multiple sclerosis, trauma, medication, noise, or prior ear surgery. All ISSHL patients were treated with the standard medical protocol, which included steroids (prednisone 1 mg/kg die with tapering), carbogen (95% $CO_2$ and 5% $O_2$) inhalation, pentoxifylline, vitamin C, magnesium sulfate.

### Anthropometric and clinical parameters

The demographic, anamnestic and clinical characteristics are shown in Table 1. Height (cm), weight (kg), waist and hip circumference and blood pressure were measured, and body mass index (BMI) (kg/m2) was calculated. A fasting blood sample was obtained for measuring plasma glucose, total cholesterol (TC), high (HDL-C) and low (LDL-C) density lipoprotein cholesterol and triglycerides (TG). For MetS we used diagnostic criteria based on IDF consensus definition, in addition each participant was considered a "current daily smoker" if they regularly smoked at least 5 cigarettes/day during the previous 3 months or had stopped smoking less than 1 year before admittance to our department [16].

### Audiovestibular investigation

All of the patients underwent a standard evaluation that consist of bed side examination, pure tone audiometry, speech audiometry, tympanometry with stapedial reflexes; auditory brainstem responses (ABR), Vestibular Evoked Myogenic Potential (VEMPS) and Magnetic Resonance Imaging (MRI) of the internal auditory canal with gadolinium. The air conduction pure-tone average (PTA) was obtained by averaging the air conduction thresholds at 0.25, 0.5, 1, 2, 3, 4 and 8 kHz. Pure-tone and speech audiometry were tested at hospitalization, at hospital discharge and 1 month after discharge. Hearing Recovery was evaluated with the Siegel's criteria [17]. Complete recovery was considered when final hearing level was better than 25dB. Patients who showed >15 dB hearing gain and whose final hearing level was between 25 and 45 dB were defined as "partial recovery". "Slight recovery" meant a final hearing level over 45 dB with hearing gain >15 dB. Patients who showed <15 dB gain was considered "no

**Table 1. Characterization of the study population.**

|  | CONTROL | ISSHL | p value |
|---|---|---|---|
| **Age (year)** | 48,34 ± 11,13 | 53,70 ± 13,73 | 0,0531 |
| **Sex** | | | |
| **Male** | 24 (55,55%) | 23 (58,97%) | 0,7548 |
| **Female** | 20 (45,45%) | 16 (41,03%) | 0,1892 |
| **Smokers** | 17 (38,64%) | 11 (18,21%) | 0,3188 |
| **BMI (Kg/m2)** | 25,23 ± 3,52 | 26,80 ± 3,44 | **0,0436** |
| **BMI classes** | | | |
| **Underweight/ Normal Weight (BMI <25)** | 21 (47,73%) | 13 (33,33%) | 0,1857 |
| **Overweight (25 ≥ BMI >30)** | 18 (40,91%) | 21 (53,85%) | 0,2413 |
| **Obese (BMI ≥ 30)** | 5 (11,36%) | 5 (12,82%) | 0,8394 |
| **Waist (cm)** | 86,51 ± 6,36 | 96,95 ± 13,77 | **<0,0001** |
| **WHR** | 0,94 ± 0,08 | 0,98 ± 0,10 | **0,0465** |
| **Glycaemia (mg/dl)** | 87,20 ± 11,85 | 101,30 ± 18,65 | **<0,0001** |
| **SBP (mmHg)** | 116,25 ± 11,11 | 131 ± 14,40 | **<0,0001** |
| **DBP (mmHg)** | 76,25 ± 7,63 | 80,1 ± 8,85 | **0,0363** |
| **TC (mg/dl)** | 188,89 ± 38,44 | 193,69 ± 37,27 | 0,5663 |
| **HDL-c (mg/dl)** | 54,61 ± 13,33 | 56,08 ± 13,19 | 0,6157 |
| **LDL-c (mg/dl)** | 113,87 ± 32,28 | 118 ± 33,80 | 0,5709 |
| **TG (mg/dl)** | 103,04 ± 69,47 | 106 ± 66,6 | 0,8439 |
| **TG-HDL ratio** | 2,25 ± 2,43 | 2,19 ± 2,04 | 0,9040 |
| **LDL-HDL ratio** | 2,20 ± 0,85 | 2,25 ± 0,89 | 0,7943 |
| **TC-HDL ratio** | 3,64 ± 1,21 | 3,65 ± 1,11 | 0,9689 |
| **AIP** | 0,23 ± 0,29 | 0,23 ± 0,31 | 1.0000 |

Data are presented as mean ± SEM (standard error of the mean). Abbreviations: Body Mass Index, BMI; Waist Hip Ratio, WHR; systolic blood pressure, SBP; dyastolic blood pressure, DBP; total cholesterol, TC; triglyceride, TG; high-density lipoprotein cholesterol, HDL-c; low-density lipoprotein cholesterol, LDL-c; atherogenic index of plasma, AIP.

improvement". In the present study as in Zhang et al [13], complete and partial recovery were defined as "*recovered*" and slight and no recovery were defined as "*no recovered*".

The study was performed in accordance with the principles of the 1983 Declaration of Helsinki. The study has been executed according to the normal clinical practice guidelines and the analysis was ex-post on data that do not interfere with patients' privacy. All the data were entered in a computerized database and were anonymized and de-identified prior to analysis. The study protocol was approved by the Ethical Committee of the Azienda Ospedaliero-Universitaria Policlinico di Bari, Italy. All patients gave their informed consent for the use of clinical data. Statistical analysis was performed using the Software R 3.5.2.

Student t-test were performed to assess comparisons among two groups in terms of continuous variables. Pearson $\chi^2$ test was used for comparisons in terms of categorial variables. P<0.05 was considered statistically significant.

## Results

### Characterization of the study population

The study group comprised 39 patients affected by ISSHL (23 males and 16 females) and 44 controls. Table 1 reports the demographic and clinical characteristics of the patients.

**Table 2. Number of patients with and without diagnosis of MetS in control and ISSHL group.**

|  |  | CONTROL | ISSHL | p value |
|---|---|---|---|---|
| NO MetS | Patient without central obesity | 19 (43,18%) | 10 (25,64%) | 0,0964 |
|  | Patients with central obesity and with < 2 MetS Criteria met | 20 (45,45%) | 14 (35,90%) | 0,3801 |
| TOTAL NO MetS |  | 39 (88,63%) | 24 (61,54%) | **0,0042** |
| MetS | Patients with central obesity and with >2 criteria met | 5 (11,36%) | 15 (38,46%) | 0,0042 |
| TOTAL MetS |  | 5 (11,36%) | 15 (38,46%) | **0,0042** |

All data are presented as number (%).

Patients affected by ISSHL presented a BMI, waist circumference, WHR (waist to hip ratio), fasting glucose and blood pressure significantly higher compared to controls.

## MetS distribution in study population

In 19 controls and 10 ISSHL patients, central obesity was in the normal value range, therefore, they did not present MetS. In 20 controls and 14 ISSHL is patients, central obesity represented a MetS criterion but they did not present other 2 MetS criteria, therefore also these patients did not have MetS diagnosis. The number of patients without a diagnosis of MetS was significantly higher in control group compared to ISSHL group. Considering patients with central obesity, 5 (11.36%) controls and 15 (38.46%) ISSHL patients met the criteria of MetS, the difference between the two groups was statistically significant (Table 2). In patients with central obesity we evaluated the number of MetS Criteria met and the means of MetS Criteria met, for each group. ISSHL patients presented a higher number of MetS criteria compared to controls. These data are showed in Table 3.

## Audiometric parameters before, at discharge and after 30 days

In the whole group before treatment average PTA on the affected ear (PTA_Pre AE) was 61.7 ± 27.9 dB HL and on the opposite ear (PTA_CL) was 23.8 ± 22.1 dB HL. Average PTA on the affected ear at discharge (PTA_D) was 47,5 ± 34.2 dB HL and 43.5 ± 33.3 dB HL after 30 days (PTA_30D). Complete recovery at 30 days was present in 19 patients (48.72%), partial recovery in 12 (30.77%), slight recovery in 2 (5.13%) and no recovery in 6 (15,38%). ISSHL patients were divided in 2 groups according to the presence of MetS. No significant differences were found in terms of pre- and post-treatment hearing both on the affected and contralateral ear (Table 4).

**Table 3. Number and mean of MetS criteria met in central obesity patients for each group.**

|  | CONTROL | ISSHL | p value |
|---|---|---|---|
| MetS Criteria met in patients with central obesity |  |  |  |
| 0 | 6(24,00%) | 4 (13,79%) | 0,3400 |
| 1 | 14 (56,00%) | 10 (34,48%) | 0,1159 |
| 2 | 3 (12,00%) | 9 (31,04%) | 0,0964 |
| 3 | 2 (8,00%) | 4 (13,79%) | 0,5036 |
| 4 | 0 (0%) | 2(6,90%) | 0,1848 |
| Mean of MetS Criteria met in patients with central obesity | 1,04 ± 0,84 | 1,66 ± 1,11 | **0,0265** |

Number of MetS criteria met are presented as Number (%). Mean of MetS criteria met are presented as mean ± SEM (standard error of the mean).

**Table 4. Audiometric parameters in ISSHL.**

|  | PTA_Pre AE | PTA_CL | PTA_D | PTA_30D |
|---|---|---|---|---|
| MetS (dB) | 61,02 ± 28,66 | 31,42 ± 18,07 | 52,79 ± 35,83 | 50,75 ± 36,51 |
| No MetS (dB) | 61,66 ± 24,71 | 22,03 ± 8,85 | 48,31 ± 29,41 | 45 ± 28,63 |
| P value | 0.9435 | 0.0693 | 0.673 | 0.6078 |

Data are presented as mean ± SEM (standard error of the mean). Abbreviations: Mean of Pure Tone Average in affected ear before treatment, PTA_Pre AE; Mean of Pure Tone Average in contralateral ear, PTA_CL; Mean of Pure Tone Average at discharge, PTA_D; Mean of Pure Tone Average after 30 days, PTA_30D.

**Table 5. Recovery rate in patients with and without MetS, according to Siegel criteria.**

|  | Total (39) | MetS (15) | No MetS (24) | P value |
|---|---|---|---|---|
| Complete Recovery | 19 (48,72) | 5 (33,33) | 14 (58,33) | 0.1336 |
| Partial Recovery | 12 (30,77) | 4 (26,67) | 8 (33,33) | 0,6652 |
| **Recovered** | **31 (79,49)** | **9 (60,00)** | **22 (91,66)** | **0.0187** |
| Slight Recovery | 2 (5,13) | 1 (6,67) | 1 (4,17) | 0,7340 |
| No Recovery | 6 (15,38) | 5 (33,33) | 1 (4,17) | 0,0154 |
| **No Recovered** | **8 (20,51)** | **6 (40,00)** | **2 (8,34)** | **0,0187** |

Data are presented as Number (%). Recovered: complete and partial recovery; No Recovered: slight and no recovery. Siegel Criteria: Complete recovery: final hearing level is better than 25dB; Partial recovery: >15 dB hearing gain and final hearing level is between 25 and 45. Slight recovery: final hearing level over 45 dB with hearing gain >15 dB. No improvement: <15 dB gain.

## Recovery rate in MetS and No MetS group

Analysys of recovery rates according to Siegel criteria confirmed that MetS had a negative association with the rate of recovery from ISSHL; in fact, a complete or partial recovery was observed in 60% of patients with MetS and in 91,66% of patients without MetS (Table 5). The number of patients who showed a slight or no recovery was significantly higher in MetS group than in No MetS group (40% VS 8.34%).

## Discussion

MetS and ISSHL are both considered risk factors for cerebrovascular accidents especially if the hearing loss is associated with vertigo [7–10]. In the present study we have evaluated the association between the presence of MetS and ISSHL and we have showed that the average of MetS criteria met by ISSHL patients were significantly higher compared to controls. In particular, compared to controls, ISSHL patients presented a significantly higher BMI, waist circumference, WHR (waist to hip ratio), fasting glucose and blood pressure. Similar results were reported by Chien et al in 2015 [12] that used, as in the present study, abdominal circumference as the main diagnostic criterion to define MetS.

Abdominal obesity has been shown to be independently associated with the other MetS components and represents by itself a risk factor for cardiometabolic diseases [11,18,19].

Abdominal obesity is associated with macrovascular and microvascular dysfunction, in particular preclinical and clinical studies have shown that obesity induce endothelial dysfunction that disrupts blood flow, reduces vascular tone and impairs the blood brain barrier [20]. The inner ear is a high metabolic organ without collateral blood supply, therefore an impaired cochlear blood perfusion mediated by endothelial dysfunction could induce hearing loss and

prevent the therapeutic agents reaching the impaired tissue through the blood circulation [4,21].

In the present study ISSHL patients presented higher levels of fasting glucose compared to controls and similar results were reported by other authors [22,23]. It has been proposed that hyperglycemia, together with insulin resistance and excessive fatty acid production may contribute to atherosclerosis onset and microvascular dysfunction with subsequent damage and apoptosis of endothelial cells [24,25] leading to sudden hearing loss.

As reported by Jalali and Nasimidoust Azgomi [26] also in the present study ISSHL patients presented higher blood tension values compared to controls. There is however limited evidence regarding the relation between hypertension and risk of hearing loss. While cross-sectional studies have shown higher prevalence of hearing loss among people with hypertension [27,28], a prospective study has suggested no association between hypertension and hearing loss [29].

In the present study we did not find higher levels of lipids in ISSHL patients. Similar results were reported by Chang et al [30], while other authors have reported a correlation between ISSHL and hyperlipidemia [31,32]. The role of hyperlipidemia needs therefore to be further investigated.

In the present study MetS was associated with a poorer hearing recovery. In particular although hearing level at admission was not different between the two groups, subjects without MetS presented a significantly higher rate of recovery compared to those with MetS.

The role of MetS as a negative prognostic factor in ISSHL has been reported by different authors [13–15], in addition hearing recovery has been negatively correlated with the single components of MetS such as diabetes [13,15] and hyperlipidaemia [13].

Zhou et al. [15], in a series of 228 patients affected by ISSHL, showed that MetS was associated with poorer hearing recovery; similar results were reported by Zhang et al [13] in 94 patients and Jung et al in 124 patients [14]. It should be noted that in all these studies obesity was evaluated with the BMI and not with the waist circumference.

The present study is therefore the first that correlates poor hearing recovery with abdominal obesity, that as previously stated, is associated with insulin resistance and endothelial dysfunction mediated by inflammation and reactive oxygen species [9].

MetS could induce therefore a subclinical damage at the level of the cochlea that not only favours the loss of hearing in case of external stressors, but also reduce the cochlear "reserve" influencing hearing recovery.

Other authors have evaluated the prognostic role of other cardiovascular risk factors in ISSHL prognosis. Hyperglycaemia has been reported as a negative prognostic factor for recovery in ISSHL [22,33], and microvascular dysfunction and hyperglycemia induced neuropathy have been proposed as the main mechanisms associated with poorer recovery. Nagaoka et al. have reported that elevation of LDL/HDL cholesterol ration and/or hypertriglyceridemia were negatively associated with auditory gain [34], similarly Lin et al. [35] showed that comorbid diabetes or hypercholesterolemia were associated with poorer recovery. Our group has recently reported that total cholesterol concentrations were a negative prognostic factor for recovery in ISSHL. The endothelial dysfunction predisposing to the development of a pro-thrombotic state at the level of the inner ear that impairs cochlear membrane functions was proposed as the responsible mechanisms [36].

## Conclusions

MetS was associated with ISSHL and this association negatively influenced the hearing recovery of these patients. Endothelial dysfunction and microvascular damage could be responsible to both the increased prevalence and poorer recovery of ISSHL.

## Supporting information

**S1 Database.**
(XLSX)

## Author Contributions

**Conceptualization:** Massimo Rinaldi, Giada Cavallaro, Nicola Quaranta.

**Data curation:** Massimo Rinaldi, Giada Cavallaro, Marica Cariello, Natasha Scialpi, Nicola Quaranta.

**Formal analysis:** Massimo Rinaldi, Giada Cavallaro, Marica Cariello, Natasha Scialpi, Nicola Quaranta.

**Writing – original draft:** Massimo Rinaldi, Giada Cavallaro.

**Writing – review & editing:** Nicola Quaranta.

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
