## [Decision Letter · Decision Letter 0]

29 Jun 2020

PONE-D-20-16260

Metabolic Syndrome and Idiopathic Sudden Sensori-Neural Hearing Loss

PLOS ONE

Dear Dr. Quaranta,

Thank you for submitting your manuscript to PLOS ONE. After careful consideration, we feel that it has merit but does not fully meet PLOS ONE’s publication criteria as it currently stands. Therefore, we invite you to submit a revised version of the manuscript that addresses the points raised during the review process.

The manuscript entitled "Metabolic Syndrome and Idiopathic Sudden Sensori-Neural Hearing Loss" tackles a relevant subject to the ENT scientific community. However, as pointed out by the reviewers, there are some concerns that need to be addressed before this manuscript is considered for publication.

Specifically:

1) NOVELTY: There are other similar studies in the literature, with larger number of patients. Bottomline - what does your study add to the already existing literature?

2) RELEVANCE: What literature gap does this study aim to fill? This matter needs to be explored in-depth in the introduction of the manuscript.

3) FOCUS: Although the authors defined a study design aimed to evaluate the correlations between cardiovascular diseases and ISSNHL, there is an extensive discussion on findings from other authors and speculation on underlying pathophysiological mechanisms. Although important and informative, I recommend including a more in-depth analysis of the authors' own findings.

4) DATA AVAILABILITY: I strongly recommend the authors to make their raw data available as a supplementary material.

We look forward to receiving your revised manuscript.

Kind regards,

Rafael da Costa Monsanto, M.D.

Academic Editor

PLOS ONE

Journal Requirements:

2. In the ethics statement in the manuscript and in the online submission form, please provide additional information about the patient records used in your retrospective study, including:  a) the date range (month and year) during which patients' medical records were accessed; and b) the source of the medical records analyzed in this work (e.g. hospital, institution or medical center name).

3. We noticed minor instances of text overlap with the following previous publication(s), which need to be addressed:

(1) https://worldwidescience.org/topicpages/m/metabolic+syndrome+involvement.html

(2) https://en.wikipedia.org/wiki/Metabolic_syndrome

The text that needs to be addressed involves the (1) Introduction (lines 53-55), (2) Introduction section (lines 58-64).

In your revision please ensure you cite all your sources (including your own works), and quote or rephrase any duplicated text outside the methods section. Further consideration is dependent on these concerns being addressed.

Reviewers' comments:

Reviewer's Responses to Questions

**Comments to the Author**

1. Is the manuscript technically sound, and do the data support the conclusions?

Reviewer #1: Partly

Reviewer #2: Partly

2. Has the statistical analysis been performed appropriately and rigorously? 

Reviewer #1: Yes

Reviewer #2: No

3. Have the authors made all data underlying the findings in their manuscript fully available?

Reviewer #1: Yes

Reviewer #2: Yes

4. Is the manuscript presented in an intelligible fashion and written in standard English?

Reviewer #1: Yes

Reviewer #2: Yes

5. Review Comments to the Author

Reviewer #1: I read the manuscript entitled “Metabolic Syndrome and Idiopathic Sudden Sensorineural Hearing Loss” with great interest.

The authors aimed to evaluate the impact of Metabolic Syndrome (MetS) and Idiopathic Sudden Sensorineural Hearing Loss (ISSHL) and the association between the presence of MetS and ISSHL.

Major issues:

Introduction

1. The pathophysiology of ISSHL remains unknown and we have a lot of widely accepted theories. In cases of ISSHL resulting from a known intravascular insult, the loss is permanent. However, the authors describe a high rate of spontaneous recovery suggesting a vascular mechanism at the basis of this disease.

2. The lack of studies on the association between ISSHL and the subsequent risk for stroke prevents from simultaneous contributions of vascular, biochemical and metabolic factors.

Methods

1. Subjects: 39 patients with ISSHL and 44 control– small size sample.

2. What comorbidities did the patients have previously? Have these variables been taken into account?

3. The authors describe that all patients underwent the same treatment protocol (steroids, carbogen inhalation, pentoxifylline, vitamin C and magnesium sulfate. Is that treatment regimen a clinical protocol for treating patients with ISSHL in your hospital?

4. It is know that standard pure tone audiometry not only provides the criteria for diagnosis of ISSHL, characteristics of the initial audiogram have prognostic value. Have these variables been taken into account?

Results

1. There are several studies in the literature who presented similar results, with larges casuistic. For example: Jung SY, Shim HS, Hah YM, Kim SH, Yeo SG. Association of Metabolic Syndrome With Sudden Sensorineural Hearing Loss. JAMA Otolaryngol Head Neck Surg. 2018;144(4):308–314. doi:10.1001/jamaoto.2017.3144.

Discussion

1. Most of the discussion only describe findings from other authors.

Reviewer #2: This manuscript entitled: "Metabolic Syndrome and Idiopathic Sudden Sensorineural Hearing Loss" is well written and addresses a topic that is being widely discussed in the literature. however, we highlight some details that need to be clarified:

1) in the introduction, the authors approach that the pathogenesis of idiopathic sudden hearing loss is controversial, however suggest only a vascular mechanism at the basis of this disease and discussed that ISSHL endothelial dysfunction has been reported "basing all the reasoning for only vascular causes and I think it is important to emphasize that the pathogenesis of ISSHL has multiple other physopathogenic processes involved.

2) The objective is very clear "we evaluated the association between the presence of MetS and ISSHL,." and the study method is all based on a study of association and not a cause and effect relationship. For this reason the statistical tests performed were all for association between two injuries, (sudden idiopathic neurosensory loss and the metabolic syndrome) whose sample is all taken in an association study. The following conclusion is not supported by the results:

"The presence of MetS increased the risk of ISSHL" or "Endothelial dysfunction and microvascular damage could be related to both increased prevalence and poorer recovery". The methodology used in this study as well as the sample size and statistical analysis does not allow us to make these conclusions. We can only conclude that there is an association between ISSHL and metabolic syndrome and that this association is negatively influenced hearing recovery in ISSHL patients.

3) In the abstract the acronym meaning BMI and WHR needs to be scoredin the abstract the acronym meaning BMI and WHR needs to be scored.

6. PLOS authors have the option to publish the peer review history of their article (what does this mean?). If published, this will include your full peer review and any attached files.

Reviewer #1: No

Reviewer #2: No

---

## [Author Response · Author response to Decision Letter 0]

27 Jul 2020

A response to reviewer file has been uploaded

---

## [Decision Letter · Decision Letter 1]

17 Aug 2020

Metabolic Syndrome and Idiopathic Sudden Sensori-Neural Hearing Loss

PONE-D-20-16260R1

Dear Dr. Quaranta,

We’re pleased to inform you that your manuscript has been judged scientifically suitable for publication and will be formally accepted for publication once it meets all outstanding technical requirements.

Kind regards,

Rafael da Costa Monsanto, M.D.

Academic Editor

PLOS ONE

Additional Editor Comments (optional):

Reviewers' comments:

Reviewer's Responses to Questions

**Comments to the Author**

1. If the authors have adequately addressed your comments raised in a previous round of review and you feel that this manuscript is now acceptable for publication, you may indicate that here to bypass the “Comments to the Author” section, enter your conflict of interest statement in the “Confidential to Editor” section, and submit your "Accept" recommendation.

Reviewer #1: All comments have been addressed

Reviewer #2: All comments have been addressed

2. Is the manuscript technically sound, and do the data support the conclusions?

Reviewer #1: Yes

Reviewer #2: Yes

3. Has the statistical analysis been performed appropriately and rigorously? 

Reviewer #1: Yes

Reviewer #2: Yes

4. Have the authors made all data underlying the findings in their manuscript fully available?

Reviewer #1: Yes

Reviewer #2: Yes

5. Is the manuscript presented in an intelligible fashion and written in standard English?

Reviewer #1: Yes

Reviewer #2: Yes

6. Review Comments to the Author

Reviewer #1: I would like to acknowledge the time and effort the authors put into writing and answering all the questions. I congratulate the authors for tackling such an important subject.

Reviewer #2: I just had doubts in the study design part that I didn't understand the context of the word “die” in “(prednisone 1 mg / kg die with tapering)”

7. PLOS authors have the option to publish the peer review history of their article (what does this mean?). If published, this will include your full peer review and any attached files.

Reviewer #1: No

Reviewer #2: No

---

## [Editor Report · Acceptance letter]

19 Aug 2020

PONE-D-20-16260R1 

Metabolic Syndrome and Idiopathic Sudden Sensori-Neural Hearing Loss 

Dear Dr. Quaranta:

I'm pleased to inform you that your manuscript has been deemed suitable for publication in PLOS ONE. Congratulations! Your manuscript is now with our production department. 

Kind regards, 

on behalf of

Dr. Rafael da Costa Monsanto 

Academic Editor

PLOS ONE